# A Review on a Deep Learning Perspective in Brain Cancer Classification

**DOI:** 10.3390/cancers11010111

**Published:** 2019-01-18

**Authors:** Gopal S. Tandel, Mainak Biswas, Omprakash G. Kakde, Ashish Tiwari, Harman S. Suri, Monica Turk, John R. Laird, Christopher K. Asare, Annabel A. Ankrah, N. N. Khanna, B. K. Madhusudhan, Luca Saba, Jasjit S. Suri

**Affiliations:** 1Department of Computer Science and Engineering, Visvesvaraya National Institute of Technology, Nagpur 440012, India; gtandel@gmail.com (G.S.T.); atiwari.rcs@gmail.com (A.T.); 2Department of Computer Science and Engineering, Marathwada Institute of Technology, Aurangabad 431010, India; mainakmani@gmail.com; 3Global Biomedical Technologies Inc., Roseville, CA 95661, USA; 4Indian Institute of Information Technology, Nagpur 440012, India; ogkakde25@gmail.com; 5Brown University, Providence, RI 02912, USA; harman_suri@brown.edu; 6Department of Neurology, University Medical Centre Maribor, 2000Maribor, Slovenia; monika.turk84@gmail.com; 7Department of Cardiology, St. Helena Hospital, St. Helena, CA 94574, USA; lairdjr@ah.org; 8Department of Neurosurgery, Greater Accra Regional Hospital, Ridge, Accra233, Ghana; drchristopher.asare@yahoo.com; 9Department of Radiology, Greater Accra Regional Hospital, Ridge, Accra233, Ghana; aaankrah@yahoo.com; 10Department of Cardiology, Apollo Hospitals, New Delhi 110076, India; drnnkhanna@gmail.com; 11Neuro and Epileptology, BGS Global Hospitals, Bangaluru 560060, India; drmadhubk@gmail.com; 12Department of Radiology, A.O.U., Cagliari 09128, Italy; lucasaba@tiscali.it; 13Stoke Monitoring and Diagnostic Division, AtheroPoint™, Roseville, CA 95661, USA

**Keywords:** cancer, brain, pathophysiology, imaging, machine learning, extreme learning, deep learning, neurological disorders

## Abstract

A World Health Organization (WHO) Feb 2018 report has recently shown that mortality rate due to brain or central nervous system (CNS) cancer is the highest in the Asian continent. It is of critical importance that cancer be detected earlier so that many of these lives can be saved. Cancer grading is an important aspect for targeted therapy. As cancer diagnosis is highly invasive, time consuming and expensive, there is an immediate requirement to develop a non-invasive, cost-effective and efficient tools for brain cancer characterization and grade estimation. Brain scans using magnetic resonance imaging (MRI), computed tomography (CT), as well as other imaging modalities, are fast and safer methods for tumor detection. In this paper, we tried to summarize the pathophysiology of brain cancer, imaging modalities of brain cancer and automatic computer assisted methods for brain cancer characterization in a machine and deep learning paradigm. Another objective of this paper is to find the current issues in existing engineering methods and also project a future paradigm. Further, we have highlighted the relationship between brain cancer and other brain disorders like stroke, Alzheimer’s, Parkinson’s, and Wilson’s disease, leukoriaosis, and other neurological disorders in the context of machine learning and the deep learning paradigm.

## 1. Introduction 

The fatality rate due to brain cancer is the highest in Asia [1]. Brain cancer develops in the brain or spinal cord [2]. The various symptoms of brain cancer include coordination issues, frequent headaches, mood swings, changes in speech, difficulty in concentration, seizures and memory loss. Brain cancer is a form of tumor which stays in the brain or central nervous system [2]. Brain tumors are categorized into various types based on their nature, origin, rate of growth and progression stage [3,4]. Brain tumors can be either benign or malignant. Benign brain tumor cells rarely invade neighboring healthy cells, have distinct borders and a slow progression rate (e.g., meningiomas, pituitary tumors and astrocytomas (WHO Grade-I)). Malignant brain tumor cells (e.g., oligodendrogliomas, high-grade astrocytomas, etc) readily attack neighboring cells in the brain or spinal cord, have fuzzy borders and rapid progression rates. Brain tumors can be further classified into two types based on their origin: primary brain tumors and secondary brain tumors. A primary tumor originates directly in the brain. If the tumor emerges in the brain due to cancer existing in some other body organ such as lungs, stomach etc., then it is known as a secondary brain tumor or metastasis. Further, grading of brain tumors is done as per the rate of growth of cancerous cells, i.e., from low to high grade. WHO categorizes brain tumors into four grades (I, II, III and IV) as per the rate of growth [2,5,6,7,8,9] (discussed later). Brain tumors are also characterized by their progression stages (Stage-0, 1, 2, 3 and 4). Stage-0 refers to cancerous tumor cells which are abnormal, but do not spread to nearby cells. Stages-1, 2 and 3 denote cells which are cancerous and spreading rapidly. Finally in Stage-4 the cancer spreads throughout the body. It is for sure that many lives could be saved if cancer were detected at an early stage through fast and cost-effective diagnosis techniques. However, it is very difficult to treat cancer at higher stages where survival rates are low. 

Brain cancer diagnosis can be either invasive or non-invasive. Biopsy is the invasive approach where an incision is done to collect a tumor sample for examination. It is considered the gold standard for cancer diagnosis where the pathologists observe various features of cells of the tumor sample under a microscope to confirm malignancy. The physical examination of the body and brain scanning using imaging modalities constitute non-invasive approaches. The various imaging modalities such as computed tomography (CT), or magnetic resonance imaging (MRI) of brain are faster and safer techniques than biopsy. These imaging modalities help radiologists locate brain disorders, observe disease progression and in surgical planning [10]. Brain scans or brain image reading to rectify disorders is however subject to inter-reader variability and accuracy which depends on the proficiency of the medical practitioner [11]. 

The advent of powerful computing machines and decreased hardware costs has led to the development of many computer-assisted tools (CAT) for cancer diagnosis by the research community. It is projected that CAT may help radiologists in improving the precision and consistency of the diagnostic results. In this study, various CAT-based intelligent learning methods i.e., machine learning (ML) and deep learning (DL) for automatic tissue characterization and tumor segmentation has been discussed. The basic objective of this paper is to highlight state-of-the-art of brain tumor classification methods, current achievements, challenges, and find the future scope. 

The paper is organized as follows: Section 2 provides an overview of the pathophysiology of brain cancer. Section 3, Section 4, Section 5 and Section 6 discuss various imaging modalities, the WHO guidelines on brain cancer grading, brain cancer tests and characterization methodologies, respectively. Section 7 briefly introduces different brain diseases and finally, Section 8 provides an overall discussion.

## 2. Pathophysiology of Brain Cancer

The pathophysiology of brain cancer is discussed here. The reasons of occurrence of brain cancer are given from the perspective of cellular architecture and its functioning within the human body.

### 2.1. Cellular Level Architecture

The cell is the basic building block of the human body. It also defines the function of each organ within the body such as oxygen flow, blood flow and waste materials management. Each cell has a central control system known as the nucleus which contains 23 pairs of chromosomes consisting of millions of genes. The instructions for these genes are contained within deoxyribonucleic acid (DNA) [12], which is like a blueprint for genes and defines their behavior. The protein of the gene is like a messenger that communicates between the cells or between the genes themselves. The message conveyed is defined by its 3D structure [13]. Genes control the continuous process of the death of unhealthy or unwanted cells besides reproduction of healthy cells. The main cause of a cancer is uncontrolled growth of cells. A mutation alters this DNA sequence, which is the root cause of malfunctioning of the genes. There are many factors involved in DNA mutations such as environmental, lifestyle, and eating habits. 

The genes responsible for cancer are divided into three categories. We introduce and define each category in detail:(i)The first category is known as tumor suppressors that controls the cell death cycle (apoptosis) [14]. This process has two signaling pathways. In the first pathway, the signal is generated by a cell to kill itself while in the second, the cell receives the death signal from nearby cells. This process of cell death is slowed down by a mutation in one of the pathways. It stops completely if this mutation happens in both pathways, leading to unstoppable cell growth [14,15]. Some examples of cell suppressor genes are RB1, PTEN, which are responsible for cell death [16].(ii)The second category of genes is responsible for the repair of the DNA. Example of DNA repair genes are MGMT and p53 protein. Any malfunctioning in them may trigger cancer. (iii)The third group known as proto-oncogenes, are in opposition to the function of the tumor suppressor genes and are responsible for the production of the protein fostering the division process and inhibiting the normal cell death [17,18]. In healthy cells, the cell division cycle is controlled by proto-oncogenes via protein signals which are generated by the cell itself or the connected cells. Once the signal is generated, it goes through a series of different steps, which is called signal transduction cascade or pathway as shown in Figure 1. This signal may be generated by the cell itself or from the nearby cells that are directly connected to it. In this pathway, many proteins are involved to carry the signal from the cell membrane to nucleus through the cytoplasm. In this process the cell membrane receptor accepts the signal and carries the message to nucleolus through various intermediate factors. Once, the signal reaches to the nucleus, the responsible genes for transcription is activated and performs the cell division task. One of the known proto-oncogenes responsible for the transcription is RAS which acts as a switch to turn ‘on’ or ‘off’ the cell division process [19]. Mutation alters its functionality which leads to transform this gene into an oncogene. In this situation the gene is unable to switch off the cell division signal and unstoppable growth of the cells may begin.

If cancer starts in the body due to any of the above-mentioned reasons, it is known as a primary tumor which invades other organs directly. If the cancer starts through blood vessels then it known as secondary tumor or metastasis [20]. Even though the secondary tumor is formed, it needs oxygen, nutrients and a blood supply to survive. Many genes exist in the body to detect these needs and start establishing a vascular network for them to satisfy their needs. This process is known as angiogenesis and is another cause of cancer explosion [21]. The genes discussed above as well as their expended form has given in Table 1.

About 15 percent of cancers worldwide are caused by viruses [22]. The viruses infect cells by altering DNA in the chromosomes which are responsible for converting proto-oncogenes into oncogenes. Only a few cancer causing viruses have been identified i.e., DNA virus and retroviruses or oncorna viruses (an RNA virus). The four basic DNA viruses responsible for human cancers are human papillomavirus, Epstein-Barr, Hepatitis B and human herpes virus. The RNA viruses which cause cancer are Human T lymphotropic type1 and hepatitis C. Several environmental factors also affect the cells. X-rays, UV light, viruses, tobacco products, pollution and many other daily use chemicals carry carcinogenic agents. Sunlight may also alter tumor suppressor genes in skin cells leading to skin cancer. Further, the carcinogenic compounds in smoke alters the lung cells causing lung cancer [23]. 

Many studies have shown that tumor cells have unique molecular signatures and characteristics [24]. Hyperplasia, metaplasia, anaplasia, dysplasia, and neoplasia are the various stages of the cells that define the cellular abnormality during microscopic analysis. Hyperplasia is the stage, where abnormal growth of the cell starts but the cell continues to appear normal. The cell first begins to appear abnormal in metaplasia. In the anaplasia state, cells lose their morphological features and are difficult to discriminate. The cell appears to be abnormal and little aggressive in dysplasia. Anaplasia is the most aggressive stage of this abnormal cell growth, where they seem quite abnormal and invade the surrounding tissues or start flowing through the bloodstream, which is one of the leading causes of metastasis [25]. The physical changes in cells due to cancer can be captured using high resolution imaging such as MRI or CT imaging, which are the focus of the next section. 

### 2.2. Relevancy between Brain Tumor and Genes 

As discussed in the last section, mutations in certain types of genes define the cancer. In various studies, some connection is found between degree of mutation in genes and type of brain tumor, which we have summarized in Table 1. Tumor protein-53 (TP53) is involved in DNA repair and initiating apoptosis. Tp53 level is found to be quite abnormal in high-grade gliomas and mutations have been found in more than 80% of tumors [26]. The retinoblastoma (RB1) gene is a tumor suppression gene. RB1 mutation is found in approximately 75% of brain tumors and it is more relevant to glioblastoma [26]. EGFR is a trans-membrane receptor in the receptor tyrosine kinase (RTK) family. Mutation in EGFR will lead to increased cell cycle proliferation and increased tumor cell survival. It is generally associated with primary glioblastomas and approximately 40% of the mutations that caused them are found within it [27]. PTEN is a tumor suppressor gene and are responsible for about 15–40% of mutations found in primary glioblastomas. The degree of mutation may be up to 80%, indifferent glioblastoma [27]. IDH1 and IDH2 are enzymes that control the citric acid cycle. Mutations in them inhibit enzyme activity. Generally, IDH1 mutation is found less in primary glioblastoma patients (5%), but more in high grade glioblastomas (70–80%). IDH2 mutations are generally seen in oligodendroglial tumors [28]. Co-deletion of chromosomes 1p and 19q is indicative of oligodendroglial lineage and mainly seen in anaplastic oligoastrocytomas (20–30%), oligoastrocytomas (30–50%), anaplastic oligodendrogliomas (60%) and oligodendrogliomas (80%). 1p/19q helps in prognosis and treatment assessment [29]. MGMT protein is another DNA repair gene, for which 35–75% abnormality is found in glioblastomas [30]. BRAF is a proto-oncogene encoded as BRAF protein, which is involved in the cell proliferation cycle, apoptosis process and treatment assessment. BRAF mutations are generally found in pilocyticastrocytomas (65–80%), pleomorphic xanthoastrocytomas (about 80%) and gangliogliomas (25%) [26]. A-Thalassemia-mental retardation syndrome X-linked (ATRX) is a gene that encodes a protein and is associated with TP53 and IDH1 mutations. It is use as a prognostic indicator when tumors have anIDH1 mutation and it distinguishes between the tumors of oligodendroglial origin [26].

## 3. Imaging Modality

Medical imaging techniques help doctors, medical practitioners and researchers view inside the human body and analyze internal activities without incisions. Cancer diagnosis, grade estimation, treatment response assessment, patient prognosis and surgery planning are the main steps and challenges in cancer treatment. There are a number of medical imaging techniques used by hospitals across the world for different treatments. The brain imaging techniques can be categorized into two types: *i.e.*, structural and functional imaging [31,32]. Structural imaging consists of different measures related to brain structure, tumor location, injuries and other brain disorders. The functional imaging techniques detect metabolic changes, lesions on a finer scale and visualize brain activities. This activity visualization is possible due to metabolic changes in a certain part of the brain which are reflected in the scans. CT and MRI are used for brain tumor analysis and are able to capture different cross-sections of the body without surgery [33,34]. 

### 3.1. Computed Tomography Imaging

In a CT scan, an X-ray beam circulates around specific part of the body and a series of images captured from various angles. The computer uses this information to create a series of two-dimensional (2D) cross-sectional image of the organ and combines them to make a three-dimensional (3D) image, which provides a better view of the organs. Positron emission tomography (PET) is a variant of CT where a contrast agents is injected into the body in order to highlight abnormal regions. CT scans are recommended by doctors in many conditions such as hemorrhages, blood clots or cancer. However, CT scans use X-rays which emit ionizing radiation and have the potential to affect living tissues, thereby increasing the risk of cancer. In one study, it is shown that the risk of radiation in CT is 100 times higher than in a normal X-ray diagnosis [35]. 

### 3.2. Magnetic Resonance Imaging

MRI is a radiation free and therefore a safer imaging technique than CT and provides finer details of the brain, spinal cord and vascular anatomy due to its good contrast. Axial, sagittal, and coronal are the basic planes of MRI to visualize the brain’s anatomy as shown in Figure 2. The most commonly used MRI sequences for brain analysis are Tl-weighted, T2-weighted, and FLAIR [36]. Tl-weighted scan provides gray and white matter contrast. T2-weighted is sensitive to water content and therefore well suited to diseases where the water accumulates inside brain tissues. T1- and T2-weighted images are also used to differentiate cerebrospinal fluid (CSF). The CSF is colorless and found in the brain and spinal cord. It looks dark in T1-weighted imaging and bright on T2-weighted imaging. The third sequence is fluid attenuated inversion recovery (FLAIR) which is similar to T2-weighted image except for its acquisition protocol. FLAIR is used in pathology to distinguish between CSF and brain abnormalities. FLAIR can locate an edema region from CSF by suppressing free water signals, and hence periventricular hyperintense lesions are clearly visible in the images.

The comparison between the above three sequences is shown in Figure 2. Diffusion-weighted imaging (DWI) [37] is another MRI sequence that helps to detect the random movements of water particles inside the brain. As the water movement becomes restricted, an extremely bright signal on the DWI is reflected, thus the DWI technique is mostly used for acute stroke detection. Perfusion-weighted MRI (PWI) highlights the specific part of the brain where the blood flow has been altered. Diffusion-tensor MRI (DTMRI) detects water motion in tissues through a microscopic image which helps during surgical removal of the brain tumor. Functional magnetic resonance imaging (fMRI) [38] is another variant of MRI that is used for measuring the changes in blood oxygenation in order to interpret the neural activity. When a certain part of the brain is more active, it starts consuming more oxygen and blood. Consequently, an fMRI maps the ongoing activity of the brain by correlating the mental process and location. Although MRI is very useful for brain image analysis, it has some limitations compared to CT. The motion artifact effect is inferior in MRI which helps in acute hemorrhage and brain injury detection, but also causes it to require a greater acquisition time than many other imaging techniques.

### 3.3. Biopsy

Biopsies are the gold standard for all cancer diagnosis and grade estimation. In a biopsy, the color, shape, and size of the cell nuclei of tumor sample are observed. This brings complexity in manual microscopic biopsy image analysis. The accuracy depends on the experience and expertise of the pathologist and therefore, computer assisted tools can help pathologist in Digital Pathological Image (DPI) analysis and may provide better results than manual approach [39]. Hematoxylin & Eosin (H&E) staining is the most commonly used method for a biopsy sample analysis. Cytopathology is used to know the cell structure, function and their chemistry. Tissue proteins are assessed by using immuno-fluorescence imaging. 

### 3.4. Hyperstereoscopy Imaging

High-grade tumors invade the surrounding normal tissues, which makes them extremely difficult to differentiate from each other through the naked eyes of surgeon (especially glioma). Incorrect resection leads to reduced survival rate of the brain cancer patients [40,41]. In this case, hyperspectral imaging (HSI) can be used. HSI is a minimally invasive, non-ionizing sensing technique. HSI uses a wider range of the electromagnetic spectrum compared to normal three channel Red, Green and Blue (RGB) image type [41], which provides detailed information about tissues in the captured scene [42]. 

Recently, scientists have proposed a novel visualization system based on HSI, which can assist surgeons to detect the brain tumor boundaries during neurosurgical procedures [40]. This model uses both supervised (SVM and KNN) and unsupervised (K-Means) machine learning techniques to differentiate cell classes such as normal, cancerous, blood vessels/hyper-vascularized tissue and background in the spectral image. The brain cancer detection algorithm is divided into off-line (training process) and in situ (online) process. In the off-line process, the samples are labeled by experts and in the in situ process, the HSI are directly acquired from the patient for real-time image analysis in the operation theater. SVM is adapted for classification during the in situ process to get a supervised classification map, while the kNN algorithm is used to find the spatial-spectral classification map. To get the final definitive classification map, image fusion is performed between spatial-spectral classification map (derived from KNN-supervised) and hierarchical K-means map (unsupervised strategy). Finally, a majority voting (MV) method is used to fuse both images for superior results. For dimensionality reduction, a principal component analysis (PCA) algorithm is adapted in the above settings.

Another study utilizing the hyperspectral paradigm is [43], where, head and neck cancer classification was done using a deep learning (DL) technique. In this study, the authors demonstrated that DL techniques have the potential to be used as a real-time tissue classifier (tissue labeling process) using HS images to identify boundaries of the cancerous and non-cancerous tissues during surgery. A CNN network was proposed consisting of six convolution layers and three fully connected layers to classify three types of classes such as head and neck tissue, squamous-cell carcinoma and thyroid cancer. The database consisted of 50 subjects. The network was trained for 25,000 iterations using a batch size of 250. Performance was evaluated using leave-one-out cross-validation protocol while computing the performance parameters giving the accuracy, sensitivity, specificity as 80%, 81% and 78%, respectively. The CNN strategy was benchmarked against conventional ML methods such as SVM, kNN, logistic regression (LR), decision tree (DT), linear discriminant analysis (LDA) demonstrating its superiority.

### 3.5. MR Spectroscopy 

MRI is able to visualize the anatomical structure of the brain, whereas, Magnetic Resonance spectroscopy (MRS) is able to detect small biochemical changes in the brain. This property is useful for the brain tissue classification in brain tumor, stroke and epilepsy. Here, several metabolites and their products such as amino acids, lactate, lipids, alanine, etc., where, the frequency can be measured in parts per million (ppm). There are unique metabolic signatures associated with each tumor type and their grades [44], therefore, the neurologist measures the changes between normal and cancerous tissues by the frequency map of ppm of each metabolite. In [45], the authors had proposed a deep learning-based model for brain tumor diagnosis using MRS imaging techniques. The authors proposed three deep models for brain tumor classification into healthy, low or high grade tissue types. In another study [46], the authors proposed a brain tumor grading method using MR spectroscopy. The proposed method showed that metabolite values/ratios could provide better classification/grading of brain tumors using, short and long echo times (TEs). A machine learning method was proposed by authors in [47] for glioma classification into benign and malignant types. Features were extracted from MR spectroscopy and then classified using popular ML methods such as SVM, random forest, multilayer perceptron, and locally weighted learning (LWL). The best performance was achieved by random forest, giving an AUC of 0.91, while a sensitivity of 86.1% was achieved using the LWL-based method.

Each imaging modality has its own merits and demerits. Occasionally we need to combine the merits of more than one imaging modality for accurate diagnosis and assessment of various severe diseases. Combining multiple image modalities is called image fusion which helps in better diagnosis than when using a single imaging technique. Image fusion improves the image quality and may reduce randomness and redundancy of the medical images. Some of the popular methods of image fusions are [48] based on morphology, knowledge, wavelets and fuzzy logic methods. 

## 4. World Health Organization Guidelines for Tumor Grading

Cancer identification and correct grade estimation are crucial part of the diagnosis process. It helps doctors decide on a personalized treatment plan which may increase the survival expectancy of the patients. Medical practitioners or histopathologists use WHO guidelines for brain tumor grading. The WHO proposed five amendments or editions since 1979 for tumor classification, presented in Table 2. In 1979, the WHO first proposed miotic activity, necrosis and infiltration for the tumor classification. In 1993, the WHO came up with another amendment, where immune histochemistry was considered for tumor assessment. After that, a genetic profile was included in the year of 2000. In the 4th amendment, a genetic profile and histological variation were combined for the tumor analysis in the year of 2007.Recently, on May 9, 2016 the WHO published an official fifth amendment to the central nervous system (CNS) tumor classification, which may precisely define the tumor cells and helps in better tumor classification [49]. All the studies have shown that tumor cells have unique molecular signatures and characteristics which define their grade and group [50]. The WHO classifies brain tumors using four basic features such as mitoses, necrosis, nuclear atypia, and microvascular proliferation [51]. The assigned grades from the least aggressive to the most aggressive (malignant) tumors are in the range of I to IV [49,50,51,52]. Grade-I cells look nearly normal and spread slowly. Grade-II cells look slightly abnormal and grow slowly and may invade nearby tissues. These are more life-threatening than Grade-I but can be cured by a suitable treatment. In Grade-III, tumor cells seem abnormal and invade the nearby healthy brain tissues. These tumors may be treated. Grade-IV cells look completely abnormal and grow and very rapidly. Eventually, it is very difficult to sub-grade tumor due to the fuzzy difference in cell structure microscopically. Therefore, grade estimation of tumor is challenging for a pathologist. 

## 5. Brain Tumor Tests

In neurological examination, the doctor asks about the patient’s health and checks vision, hearing, alertness, muscle strength and reflexes. The doctor may also examine the eyes of a patient to see any swelling. Brain scans, tumor biopsy and biomarkers are major tests to confirm cancer and its grade. If the doctor finds any symptoms of brain cancer then they may suggest any one of them depending on the patient condition to confirm the malignancy of the brain tumor. Some of the tests are given in the following subsections.

### 5.1. Biomarker Test 

Mutation in the genes is the root cause of cancer and the degree of this mutation in specific genes can be measured through biomarker tests. Some of the genes responsible for specific brain cancers are given in Table 1. This test diagnoses tumors, helps to find its type and may help in tumor growth measurement, treatment response and personalized treatment therapy [53].

### 5.2. Biopsy 

Biopsy is the primary test for diagnosis and stage conformation [54] for all types of cancer. This is an invasive cancer diagnosis approach. In this test, a sample of the brain tumor is taken out through surgery and the procedure may take several hours. The collected biopsy samples go through a laboratory test where the histopathologists look for the cellular patterns and characteristics to estimate the grade of the brain tumor. The low and high-grades of tumor are difficult to differentiate as their cellular structures are similar. Accurate diagnosis is an important step to analyze the behavior of the tumor and make the correct treatment plan. The estimation of the grade of the tumor is subject to inter-reader variability and correct analysis of the DPI depends on the training and experience of the histopathologists [55]. Image features that grade tumors are not always clear or difficult to determine by different observers. The computerized image analysis can partially overcome these shortcomings [56]. Complexity in clinical features representation, large size single histopathology image and insufficient images for training are the major barriers in the automatics classification techniques development [56]. Computerized image analysis include image registration, preprocessing, feature selection, the region of interest (ROI) identification, segmentation and image classification which are discussed later.

For many years, The Medical Image Computing and Computer Assisted Intervention (MICCAI) Society has been organizing many conferences and open challenges that foster to develop computer assisted tools or medical inventions in medical image analysis. Recently, many digital histopathology image analysis challenges were organized worldwide to boost the tumor histopathology among researchers community. We have summarized some of the MICCA challenges in Table 3.

### 5.3. Imaging Test 

Imaging modalities such as CT, MRI, PET, and SPECT are popular brain imaging techniques to confirm the presence of tumors without using surgery. Amongst them, MRI is the most popular diagnostic imaging modality. MRI is mainly used for neural disorder or abnormality detection because of its good contrast resolution for different tissues and lack of radiation. Automatic brain tumor detection and classification is a challenging task due to overlapping intensities, anatomical inconsistency in shape, size and orientation, noise perturbations and low contrast of images [63]. Some of the open challenges proposed worldwide for brain image analysis have been summarized in Table 4. Our main focus of this review is to highlight the challenges involved and find the future scope in a non-invasive procedure of brain tumor detection and classification using the ML and DL approaches. In the next section, we have discussed various ML and DL methods for the brain image segmentation, tumor detection, and classification and point out limitations and future scope for the enhancements. 

## 6. Classification Methods

Machine learning can be defined as a situation where a machine is given a task in which the machine performance improves with experience [73]. ML algorithms are divided into two types: supervised learning and unsupervised learning [74,75]. In supervised learning, ML algorithms learn from already labeled data. In unsupervised learning, the ML algorithms try to understand the inter-data relationship from unlabeled data. In the case of brain image analysis, ML has been used in characterizing brain tumors [75,76]. The inner workings of ML algorithms consist of two stages: feature extraction and application of ML algorithm for characterization. The process model is shown in Figure 3.

The feature extraction algorithms are generally mathematical models based on various image properties such as texture, brightness, contrast. Sometimes, several features from different extraction models are fused together to increase the discrimination power of ML algorithms [77]. Some of the most common algorithms for classification and segmentation of brain images are: K-Nearest Neighbors (KNN) [78], Support Vector Machines (SVM) [79], Artificial Neural Networks (ANN) [80] etc. The KNN classification is based on the premise that features of the same class cluster together. The KNN assigns an unknown instance the most common label amongst its K nearest neighbors. The SVM applies two approaches for characterization: at first it tries to find the largest separating hyper-plane between two classes. In the second approach, if the features are not separable in one dimension, they are mapped to higher dimension where they are linearly separable, by using the kernel approach. ANN forms hierarchical network of computing nodes capable of learning from features. ANNs are classified into many types depending on their architecture, number of hidden layers, connection weight updating algorithms, etc. The most common ANN models are extreme learning machines (ELMs) [81], recurrent neural networks (RNN) [82], restricted Boltzmann machine (RBN) [83] etc. ELM is single-layer feed-forward neural network (SLFFNN), RNNs apply feedback mechanism in the network connections and RBN is a stochastic neural network.

The advent of high performance computers, as well as lower hardware costs have led to the emergence of models with multiple layers of abstraction and millions of computing nodes which has enabled characterization/segmentation with a high degree of accuracy. These models are collectively called DL methodologies [84]. The most common DL models for brain image characterization are convolution neural networks (CNN) [85], auto encoders [86] and deep belief networks (DBNs) [87]. DL-based tools for brain images are rapidly finding interest amongst the research community.

### 6.1. Machine Learning 

KNN, SVM, DT, the naive Bayes (NB) classifier, expectation maximization (EM), random forest (RF) etc. are the most popular ML techniques for medical image analysis. Many of them were used alone or in combination by various researchers for brain image analysis. Some of them are discussed in Table 4. We provide different brain cancer classification techniques using ML in the following subsections. 

#### 6.1.1. ANN-Based MRI Brain Tumor Classification Using Genetic Features

The artificial neural network (ANN)-based approach for brain tumor classification using MRI was proposed in [63]. The method is able to characterize normal (N), benign (B) and malignant (M) tumor. The N, B and image example is shown in Figure 4. 

For the purpose of characterization, 100 brain MR images (*N* = 35, *B* = 35, *M* = 30) were collected. A semi-automatic method was applied to extract the region-of-interest (ROI). A wavelet-based feature selection was performed to extract the features. A genetic-based feature selection algorithm along with principal component analysis (PCA) and classical sequential algorithm was applied for feature selection. Finally, all the features are input into the ANN. The ANN classifier is a three-layer feed forward neural network with a single hidden layer. The process model of the approach is shown in Figure 5. It’s found that the genetic approach using only four of the available 29 features attained a classification accuracy of 98%. Similar approaches such as PCA and other classical algorithms required a large feature set to achieve a similar accuracy level.

#### 6.1.2. A Hybrid Characterization System for Brain Cancer Tumors

In [88], a hybrid system consisting of two ML algorithms has been proposed for brain cancer tumor characterization. A total of 70 brain MRI images (abnormal: 60, normal: 10) were considered for this purpose. The features were extracted from the images using DWT [89]. The total numbers of features were reduced using PCA [90]. After feature extraction, two classifiers were used separately on the reduced features (i.e., feed forward back propagation based artificial neural network (FP-ANN) and KNN). FP-ANN applies to the back-propagation learning algorithm for weight updating [91]. KNN is discussed earlier. This method achieves 97% and 98% accuracy using FP-ANN and KNN, respectively. The process model of the proposed method is shown in Figure 6.

#### 6.1.3. A Characterization System for Grading Brain Cancer Tumors

A fully automated brain tumor classification scheme using conventional MRI and rCBV maps calculated from perfusion MRI was proposed in [92]. The method classifies meningioma, glioma grades (II, III, IV), and metastasis brain images as shown in Figure 7. Earlier, researchers used linear discriminant analysis (LDA) as a model based on principle component regression (PCR) [93]. In this method, a linear SVM model is used for characterization. A total of 102 MRI brain scans were used for the purpose of characterization. The images were pre-processed and ROIs were extracted. Several features were extracted such as tumor shape characteristics, image intensity characteristics and Gabor features. In order to reduce the features, selection algorithms were applied (i.e., Ranking-based and SVM-recursive feature elimination (SVM-RFE)). Finally, SVM is applied. A process model of the methodology is shown in Figure 8. The highest classification accuracy obtained for metastasis was 91.7%, while for low-grade gliomas it was 90.9%. The highest accuracy of 97.8% was achieved when distinguishing grade II gliomas from metastasis. The lowest accuracy of 75% is obtained when distinguishing grade II from grade III gliomas. This showed that grade II and III gliomas are difficult to distinguish.

#### 6.1.4. A Multi-Parametric Tissue Characterization System for Brain Neoplasm

A characterization system was developed for identifying neoplastic tissue from healthy tissue, as well as the classification of different tumor components and edema-like areas [94]. Data was collected from 14 patients recently diagnosed with brain cancer. The images were pre-processed and voxel-wise intensity feature vectors were collected. Bayesian [95,96,97] and SVM were used to distinguish neoplastic tissue from healthy tissue, as well as the classification of different tumor components and edema-like areas. The results show that the Bayesian classifier obtains higher accuracy for classifying edema, enhancing neoplasm and non-enhancing neoplasm at 97.03%, 96.39% and 93.05%, respectively. SVM obtained highest accuracy for cerebrospinal fluid at 91.34%. The process model is shown in Figure 9.

#### 6.1.5. Extreme Learning Machine

Extreme learning machine (ELM) is another emerging area which is less computationally expensive compared to neural networks. It is based on the single-layer feed-forward neural network (SLFFNN) which is used for real-time classification or regression. ELM chooses randomly initialized weights in the input-to-hidden layer, whereas, hidden-to-output layer weights are trained using Moore-Penrose inverse form [97] to generate least square solution. This feature minimizes network complexity, training time, learning speed, and improves classification accuracy. Moreover, the weights in the hidden layer give a multi-tasking capability to the network as in other ML methods like SVM, KNN and Bayesian network. The ELM network consists of three layers as shown in Figure 10 and all the layers are fully connected. The weight between input and hidden layer are fixed at random initially and unchanged throughout the training process and weights between hidden and output are only allowed to change. Therefore it learns the weights in a single pass and reaches a global optimum [98]. There is a claim of researchers [98,99] that due to its simpler architecture and one shot training makes this network better and faster as compared to SVM. 

### 6.2. Deep Learning 

DL is most extensively used for the brain image analysis in several applications such as normal or abnormal brain tumor classification, segmentation (edema, enhancing and non-enhancing tumor region), stroke lesion segmentation, Alzheimer diagnosis, etc. A convolution neural network (CNN) is the most popular DL model used widely for classification and segmentation of medical images. The CNN learns the spatial relationship between pixels in a hierarchical manner. This is done by using convolving the images using learned filters to build a hierarchy of feature maps. This convolution function is done in several layers such that the features obtained are translation and distortion invariant resulting in high degree of accuracy. The basic layers of CNN network are described below.

#### 6.2.1. Input Image Format 

The input image is considered as an array of pixel values which depends on the resolution and size of the image. For example, a sample colored input image is represented by a 3 × m × n array of numbers (the 3refers to red, green and blue color values in case of color image with the pixel value for each color ranging from 0–255; m and n are the dimensions of the image). In the case of a grayscale image, the image size is defined by 2D array (m × n), where the intensity of the pixels also ranges from 0–255.

#### 6.2.2. Convolution Layer

The first layer of CNN architecture is the convolution layer, which extracts features from the given input image using the convolution filters. The filter is a square array of numbers which are weights or parameters. These filters can loosely be thought of as the neurons of an ANN or the kernel. The first position of the filter corresponds to the top left corner of the image in the convolution operation. This operation is described in Equation (1), which shows an example of an image (*R*) being convolved with the kernel (*S*), where ⊗ denotes the convolution operation. Essentially operation can be thought of as a series of multiplications of the image pixel matrix and the filter matrix and then a summing of these multiplications. Important to note in Equation (1) is that the kernel is size of m *×* m and the operation is performed at the center pixel *(x, y)*, and nearby, where the *p* and *q* are the dummy variables. This process repeated by sliding filter to the right. The number of cell shifts to the right in each step defines the stride (number of cells sliding right in each step). The CNN architecture is shown in Figure 11. CNN learns and updates filters or kernel values during the training.
(1)$$f(x,y)=R(x,y)⊗S(p,q)=\sum_{p=-m/2}^{m/2}\sum_{q=-m/2}^{m/2}R(x+p,y+q)\times S(p,q)$$

#### 6.2.3. Activation Function

In ANNs, the training progress is measured by gradient-based methods where the gradient is considered as a learning parameter, which reflects the changes in the training process. Since the changes in gradient are very small during training then learning is not effective and this phenomenon is known as vanishing gradient problem. This problem is more severe in DL because of large number of layers. It can be avoided by using suitable activation function which, don’t have this property of suppressing the input space into a small region. ReLu is very simple and computationally inexpensive activation function which performs the non-linear operation and replaces all negative values in the feature map by zero using a simple formula [max (0, x)], whereas, x is an input parameter [100].

#### 6.2.4. Pooling Layer

To make the method computational inexpensive, a pooling layer is introduced between convolution layers to reduce the dimensionality of each feature maps but retain the most important feature information. Average pooling and max-pooling are the two popular pooling operations. In average pooling; selected patch features are replaced by the single average value of patch in next layer, whereas, for max pooling only maximum value of patch features move further.

#### 6.2.5. Fully Connected Layer 

The first three operations i.e., convolution, ReLu, and pooling are used for extracting high-level image features. For features classification, a fully connected network appended at the end of the CNN, which convert last 2D layers into a one-dimensional feature vector. The output of the FC layer defines by N-dimensional vector which refers to the number of output classes. Only one of the output class chosen from the vector by using probabilistic methods such as softmax.

### 6.3. Brain Image Analysis Using Deep Learning

As discussed earlier, DL algorithms are used in brain image analysis in different application domains like Alzheimer’s disease identification, segmentation of lesion (e.g., tumors, white matter lesions, lacunes, micro-bleeds) and brain tissue classification [101]. Much of the ongoing research is limited to brain segmentation and only little work has been done for the tumor grading. Hence, there are a lot of potentials to explore the grade estimation for brain tumor using ML and DL approaches. In this section, we have discussed some recently existing DL based brain image segmentation methods.

#### 6.3.1. DL-Based Inter-Institutional Brain Tumor Segmentation

A CNN-based brain tumor segmentation method was proposed in [102]. In the experiment, three CNNs were used for training on multi-institutional data. Each CNN consisted of four convolution layers followed by two fully connected layers. Data of 68 patients were collected from two institutes. Patching-based segmentation was used. The equal sized patches extracted from images were annotated into three classes: tumor patches, healthy patches surrounding the tumor and other healthy patches. The tumor images were further divided into five classes based on patient data i.e, class-0: normal, class-2: enhancing region, class-3: necrotic region, class-4: T1-abnormality, class-5: FLAIR abnormality, class-1: ground truth region based on combination of classes 2–5. The various classes of tumor are shown in Figure 12.

The first CNN was trained for the institution-1 data set, second for the institute-2 dataset and third CNN was trained for patients from both institutions. Dice similarity coefficients and Hausdorff distance were used for the assessment between the ground truth and automatic segmentation. Ten-fold cross-validation scheme was applied to compare the performance between different approaches. They observed that performance of the model decreased when network is trained and tested on different institutional data (dice coefficients: 0.68 ± 0.19 and 0.59 ± 0.19) in comparison with same institutional data (dice coefficients: 0.72 ± 0.17 and 0.76 ± 0.12) and concluded that the reasons behind this effect require extra comprehensive investigation. The process model is shown in Figure 13.

#### 6.3.2. Brain Tumor Segmentation Using Two-Pathway CNN

Two-pathway based fully automated segmentation method was proposed for brain tumors [103]. The method segments glioblastomas (low grade glioma/LGG and high grade glioma/HGG) from MR images. The two pathways are executed using a small convolution filter for local segmentation and large filter for global segmentation. At last the feature maps from both pathways are concatenated to give us the segmented image. Based on this approach three cascaded networks were developed: Input Cascade CNN, MF Cascade CNN and Local Cascade CNN. The Input Cascade CNN obtained the highest Dice similarity of 0.89. The segmented results are shown in Figure 14. The architecture of the model is shown in Figure 15.

### 6.4. Plausible Solution for Brain Cancer Classification

Gliomas are the most common brain tumor in adults, and are generally divided into two categories: HGG and LGG. The WHO further divides LGG into I-II grade tumors and HGG into III-IV grade. Features such as shape and size of cell and its nuclei and cellular distribution are used to measure the degree of malignancy of the tumor microscopically. Differentiating HGG and LGG is somewhat easier than further sub-classification between LGG grade-I and II or HGG grade-III and IV, due to their uneven structure of the cell in this state. Grade estimation of the cancer is a very important parameter to decide targeted therapy and assessment of prognosis. Although biopsy is the gold standard, it is inherently invasive, along with its sampling errors and variability in interpretation, therefore, most doctors prefer MRI (T1, T2, and FLAIR) test in case surgical resection is difficult due to the location of tumor or patient condition, because of its good contrast and radiation-free nature from brain scans (MRI, CT, etc.). Most of the medical practitioners manually measure the degree of aggressiveness (grade) of the tumor. The accuracy of grade estimation depends on the proficiency of the practitioners and subjected to inter-reader variability studies. In this case, computer-assisted tools may help for better accuracy.

There are some automatic brain tumor grading methods which were proposed by researchers based on texture analysis using ML techniques [92,104,105]. Most of them use MRI (T1, T2, FLAIR, etc.). Recently many DL architectures (especially CNN) have shown remarkable performance in medical image analysis such as brain tumor segmentation and tissue classification on brain MRI. However, tumor grading utilizing DL methods is unexplored so far and there is a lot of research scope to explore further. We have provided a plausible solution for the tumor grading as shown in Figure 16. The model is described vividly in the discussion section.

## 7. Brain Cancer and Other Brain Disorders 

### 7.1. Stroke 

There are two major classes of stroke: ischemic and hemorrhagic stroke [106]. Ischemic strokes happen when blood supply is interrupted in the brain, while hemorrhagic strokes results from blood vessel damage or abnormal vascular structure. Although stroke and brain cancer are two different diseases, the relationships between them have been examined by some researchers. A study was done on longitudinal risk of developing brain cancer in stroke patients [107]. For this study, they have selected 35 cases of malignant gliomas with or without stroke cases using brain MRI. They observed that the stroke patients have a higher risk of developing brain cancer than other forms of cancers with a hazard ratio of 3.09 (95% Confidence Interval (CI): 1.80–5.30). Another interesting finding of the study is that the old stroke patients and females between 40–60 age groups have more risk of developing brain cancer. 

### 7.2. Alzheimer’s Disease 

Alzheimer’s disease (AD) is a chronic neurodegenerative disease, where the short term memory loss is an initial symptom which may become worse over the time as disease advances i.e., language problem, behavioral issues, and the inability of self-care, etc [107]. Although, AD and cancer are two different diseases there is relationship between them in some studies. It is found that there is an inverse relationship between cancer and Alzheimer’s disease in their study. Over a mean follow-up of 10 years of patients, they found that the cancer survivors have a 33% decreased risk of Alzheimer’s disease as compared to the people without cancer. Another interesting outcome came out of the study is that the patients who have AD had risk of cancer decreased by 61%. 

### 7.3. Parkinson’s Disease 

Parkinson’s disease (PD) mainly affects the motor system of the brain resulting in tremors, rigidity, and slowness in movement and difficulty in walking. Sometimes thought process and behavioral changes are also observed [108]. A meta-analysis for demonstrating the relationship between PD and brain found a positive connection between them. Eight groups were involved in the study where 329,276 patients had participated. The study revealed that occurrence of brain tumor was relatively higher after the diagnosis of PD (odds ratio 1.55, 95% CI 1.18 ± 2.05), but not statistically significant before PD diagnosis (odds ratio 1.21, 95% CI 0.93 ± 1.58).

### 7.4. Leukoaraiosis

Leukoaraiosis is an abnormal change in the appearance of white matter near the lateral ventricles. It is often seen in old age, but sometimes also found in young adults. Leukoaraiosis may be the initial stage of Binswanger’s disease but this may not always happen [109]. We cannot find any direct relation between brain cancer and Leukoaraiosis. 

### 7.5. Multiple Sclerosis 

Multiple sclerosis (MS) is a brain and spinal cord disease. In this disease, the immune system attacks the protective sheath (myelin) that covers nerve fibers which hampers communication system from the brain to rest of the body. The severity of the disease is measured by the quantity of nerve damage. Signs and symptoms of the disease may differ person to person. The symptoms are partial or complete loss of vision, double vision, speech slur, tingling in different parts of the body and losing walking ability at a higher stage. There is no permanent cure available for MS. In a recent study, it was shown that the MS patients have an increased risk of brain cancer [110,111].

### 7.6. Wilson’s Disease

Wilson’s Disease (WD) is caused by genetic disorder which is inherited from the parents. In this disease, copper builds up in the body and generally affects the brain and liver. Vomiting, weakness, fluid buildup in the abdomen, swelling of the legs, yellowish skin, and itchiness are some common liver related symptoms. Brain-related symptoms are tremors, muscle stiffness, trouble in speaking, personality changes, anxiety and seeing or hearing things [112]. A comparison of the differences in brain diseases is shown in Figure 17.

## 8. Discussion 

Brain tumor analysis using medical imaging is a complicated and challenging task, which can be broadly categorized into pre-processing, classification and post-processing steps. There are many challenges associated with the aforementioned steps, which make this task complicated. No ideal computer assisted tools available so far to conform, tumor malignancy and its degree of aggressiveness. Thus doctors rely on the biopsy test [54,55] only for all types of cancer. The manual microscopic biopsy image analysis is done by pathologists and medical practitioners by observing cell or tissue structure under the microscope. The analysis is a challenging issue for them and subject to inter-reader variability tests. Therefore, DPI analysis is a growing area of research. In DPI, some common features include the shape and size of cells, shape and size of cell nuclei and distribution of the cells which are used to measure the degree of malignancy of the tumor. Characterizing benign and malignant cells is easier than sub-classifying malignant tumor due to uneven structure of the cell in this state. Staining variations, usage of different scanners and colors variations of the tissues may appear in DPI which may lead to wrong interpretation. Another challenge with DPI is that most of the whole slide image (WSI) scanner generates only 2D image, whereas the depth information is unavailable in 2D image, which is an important parameter for pathologists to confirm certain tissue class. It is anticipated that the design of 3D WSI scanners may be available soon [120]. Since biopsies are time-consuming and more risk-prone in the case of the brain tumor, therefore, various brain scans such as CT, MRI, etc. are used to confirm tumors and the degree of malignancy. Again, this analysis depends on the proficiency of the medical practitioners and is subject to inter-reader variability. 

As discussed above, many automatic brain image analysis methods were proposed by various researchers for brain segmentation and tissue classification. Most of them use MRI (T1, T2, and FLAIR), due to its good contrast and radiation-free nature. As discussed earlier, brain image analysis consists of image registration, image enhancement, features reduction, feature extraction and classification. The image registration is the first and most important step in medical imaging. Image acquisition is not always consistent because of the effects of noise and blurring due to organ movements. The performance of the medical image analysis highly depends on several parameters such as modality, similarity measures, transformation, image contents, optimization of algorithm and implementation mechanism. Generally medical images suffer from low contrast which leads to deterioration of image quality. Gaussian (high-pass, low-pass) filter, histogram equalization, contrast starching are most commonly used image enhancement techniques for medical images. Large numbers of features are computationally expensive and make classification complex. Therefore, principal component analysis (PCA), linear discriminant analysis (LDA), and genetic algorithm (GA) are the most popular methods for feature reduction. SVM, DT, naive Bayes classifier, Bayesian classifier, KNN, ANNs etc. are the most commonly used ML methods for brain image classification and have achieved high-level accuracy in classification. In ML, features are first extracted by using hand-made techniques and then input to the ML-based characterization system. The difficulty of image classification using ML-based algorithms is that there lies continuous variability within image classes. Further, the contemporary distance measures used by feature extraction methods are unable to compute similarity between images. Nowadays, DL methods (CNN’s, ResNets) are gaining more popularity than ML techniques for the brain image classification. In DL, the images are directly input to the system. DL models such as CNN produce features from images which are translation invariant and stable to deformations leading to more accurate characterization/segmentation. In addition to characterization/segmentation of brain, it is suggested to utilize DL models for grading of brain tumor. A proposed DL-based model is already shown in Figure 16. There are four CNNs (CNN-1, 2, 3 and 4) employed for brain cancer characterization and grading. Brain MR Images are first pre-processed and tumor part is segregated. The tumor part is characterized as normal, benign or malignant. If the tumor is malignant CNN2 is employed to characterize it as LGG or HGG. LGG is further characterized as tumor grade-I or grade-II using CNN3. Similarly, HGG is classified as tumor grade-III and grade-IV by CNN4. This model can effectively diagnose brain cancer and do its grading. 

Although DL methods are widely popular among the research community, there are many challenges involved with DL architectures. DL models are quite computationally expensive because of additional hardware (GPUs) requirements to run the models. The memory and processing requirement of DL models are huge. It is also not necessary that increasing the number of layers in DL architecture will improve the performance of the architecture.

### 8.1. A Note on Biomarkers for Cancer Detection

Various tests have been suggested for diagnosing brain cancer: (a) including the one stated earlier in the section of imaging modalities, such as MRI, MRS, CT, etc., and (b) laboratory sampling of brain tumor i.e., biopsy. The inclusion of intelligence-based techniques such as ML or DL for imaging modalities are very likely to increase the effectiveness of the diagnosis and enhance the radiologists’ capability towards accurate diagnosis for brain cancer in a timely manner. In addition to the computer-aided diagnosis using imaging modalities and biopsy methodologies, spread of cancer in the nervous system can be detected using a sample of cerebrospinal fluid from the spinal cord. This technique is called lumbar puncture or spinal tap [121]. In this methodology, several biomarkers related to brain tumor were detected [122]. In addition, molecular tests on brain tumor sample can be carried out to identify specific genes, proteins, and cells related to the particular tumor. Doctors can look into these biomarkers to assess the grade, type of tumor and decide treatment options. Further, examining these biomarkers can help in early treatment before the symptoms begin. Inclusion of ML and DL techniques in assessing these biomarkers can lead to accurate diagnosis that can save both time and cost, proving to be more economical.

### 8.2. Benchmarking

The benchmarking of several ML-based brain cancer classification system has been provided in Table 5. Sasikal et al. (Row #1) applied ANN-based classifier on featured extracted using DWT from 100 T2W MRI images. The accuracy obtained is 98%. In 2008, Verma et al. (Row #2) applied Bayesian and SVM on 14 DWI, B), FLAIR, T1 and GAD images and achieved sensitivity of 91.84% and specificity of 99.57% for SVM. Zacharaki et al. achieved 97.8% accuracy using NL-SVM on SVM-RFE features from 102 T1,2 FLAIR, rCBV images (Row #3). EL Dahashanet al. (Row #4) in 2009, applied FP-ANN and KNN on features extracted using DWT from 70 MR images and obtained highest accuracy of 98.0%. Similarly, Ryu et al. (Row #5) applied entropy histogram techniques on GLCM features extracted from 42 DWI, ADC images and achieved accuracy of 84.4%. Further, Skogen et al. (Row #6) applied standard deviation on 95 patients from 95 T1W, T2 and FLAIR images and also achieved an accuracy of 84.4%.

## 9. Conclusions

Our main focus of the review is to provide state of art in brain cancer area that includes pathophysiology of cancer, imaging modality, WHO guidelines for tumor classification, primary diagnosis methods, and existing computer-assisted algorithms for brain cancer classifications using the machine and deep learning techniques. Finally, we have compared brain tumor with other brain disorders. We have concluded that due to automatic feature extraction capability of DL based methods, recently it is getting more attention and accuracy compared to conventional classification techniques for medical imaging. It is for sure that many lives can be saved if cancer detected and suitable grade estimated through fast and cost-effective diagnosis techniques. Therefore, there is dare need to develop fast, non-invasive and cost effective diagnosis techniques. Here, DL methods can play a major role for the same. In best of our knowledge, very less work has done for the automatic tumor grading using DL techniques and their full potential, yet to be explored.

## Figures and Tables

**Figure 1 cancers-11-00111-f001:**
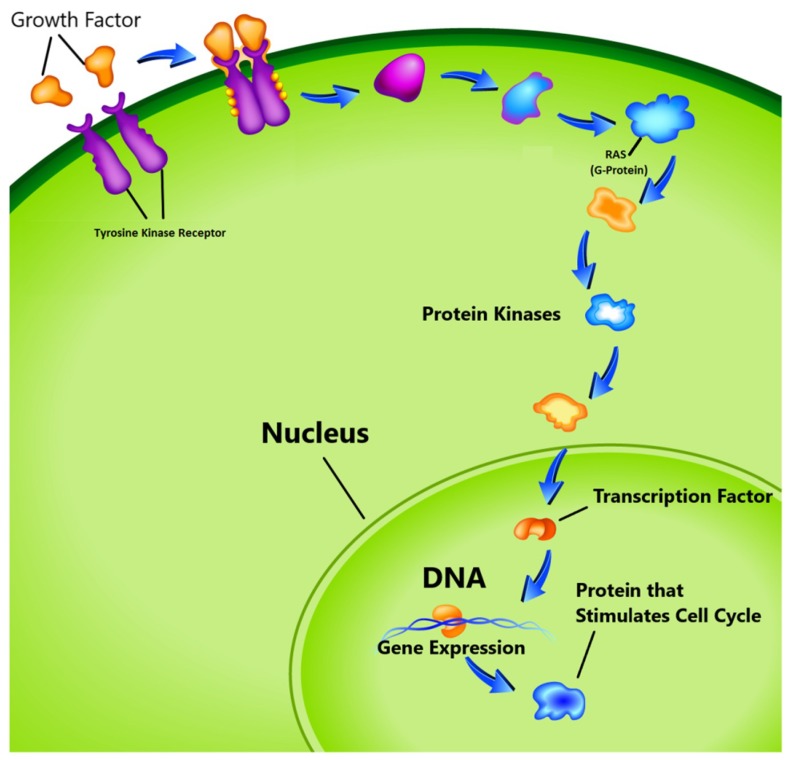
Cell cycle proliferation. (image courtesy: AtheroPoint^TM^, Roseville, CA, USA).

**Figure 2 cancers-11-00111-f002:**
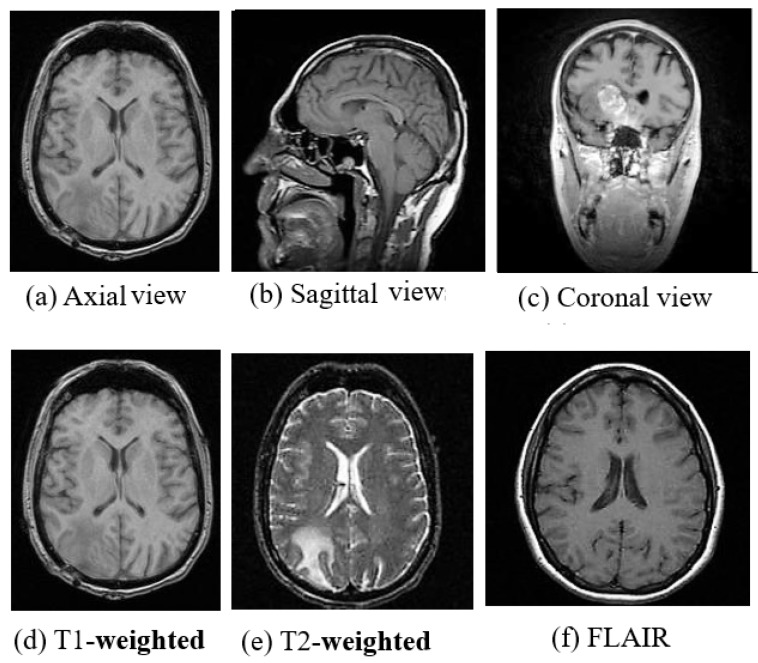
(**a**) Axial view, (**b**) Sagittal view, (**c**) Coronal view and (**d**) T1-weighted, (**e**) T2-weighted and (**f**) FLAIR Images of MRI. (image courtesy: AtheroPoint^TM^ ).

**Figure 3 cancers-11-00111-f003:**
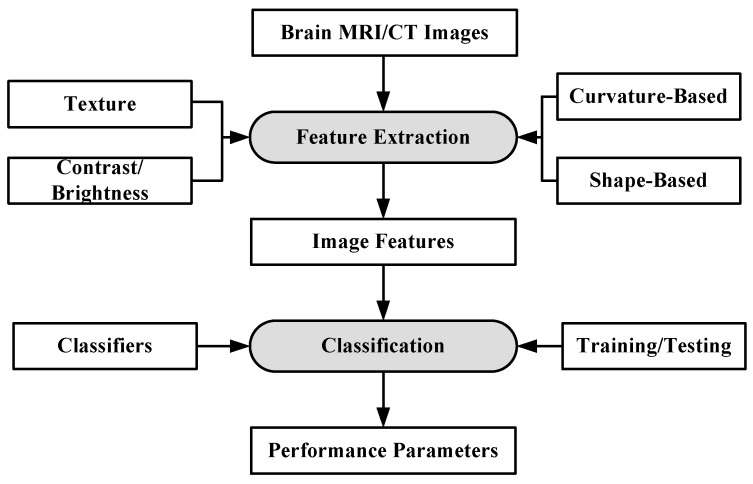
Working of ML-based algorithms.

**Figure 4 cancers-11-00111-f004:**
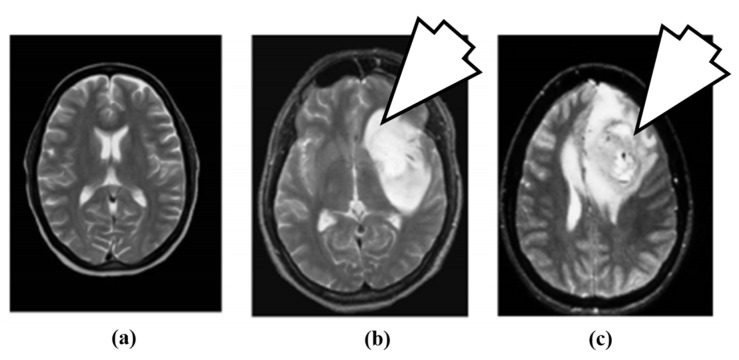
Brain MR images: (**a**) normal brain, (**b**) benign tumor (7 O’ clock arrow) and (**c**) malignant tumor (7 O’ clock arrow) (reproduced from [63] with permission).

**Figure 5 cancers-11-00111-f005:**
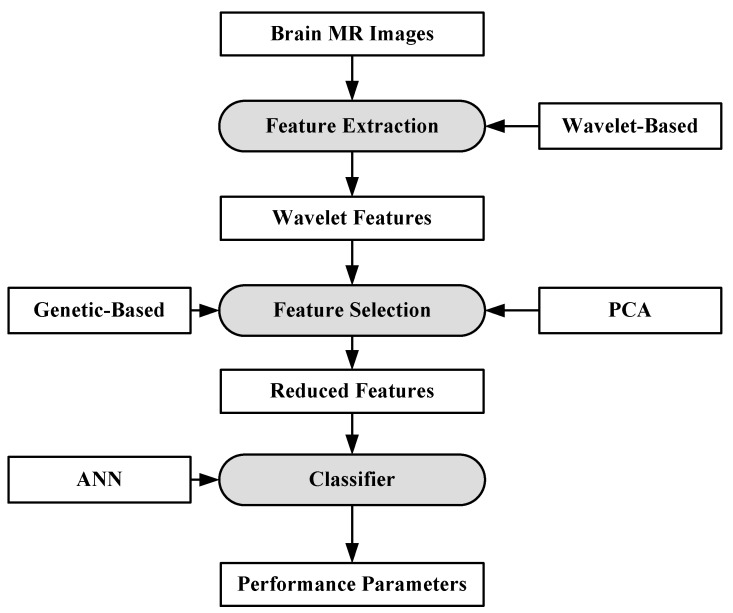
Process model of ANN-based classification model [63].

**Figure 6 cancers-11-00111-f006:**
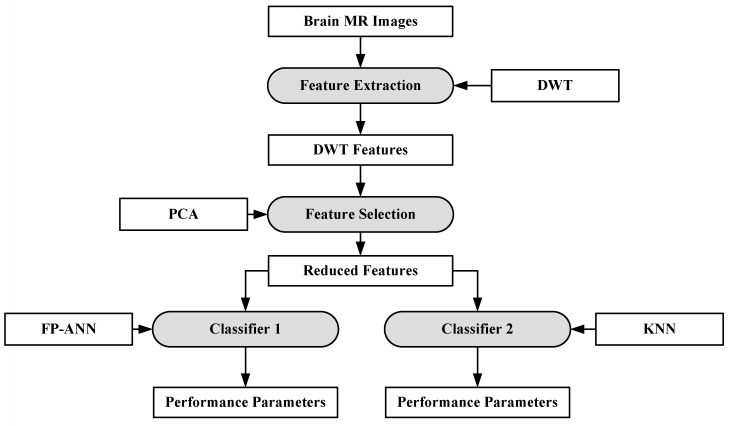
Hybrid characterization system for brain cancer characterization [88].

**Figure 7 cancers-11-00111-f007:**
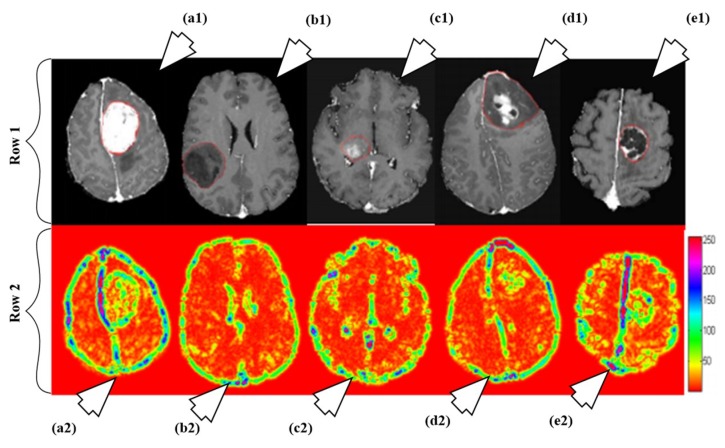
Illustration of different types as per their grades: row 1 and row 2 consists of T1ce brain images and its corresponding texture images, respectively. The images are pointed to by arrow are as follows: a1 (T1ce) and a2 (Texture): meningioma; b1 (T1ce) and b2 (Texture): Grade-II; c1 (T1ce), c2 (Texture): Grade-III; d1 (T1ce) and d2 (Texture): Grade-IV; e1 (T1ce) and e2 (Texture): metastasis (reproduced from [92] with permission).

**Figure 8 cancers-11-00111-f008:**
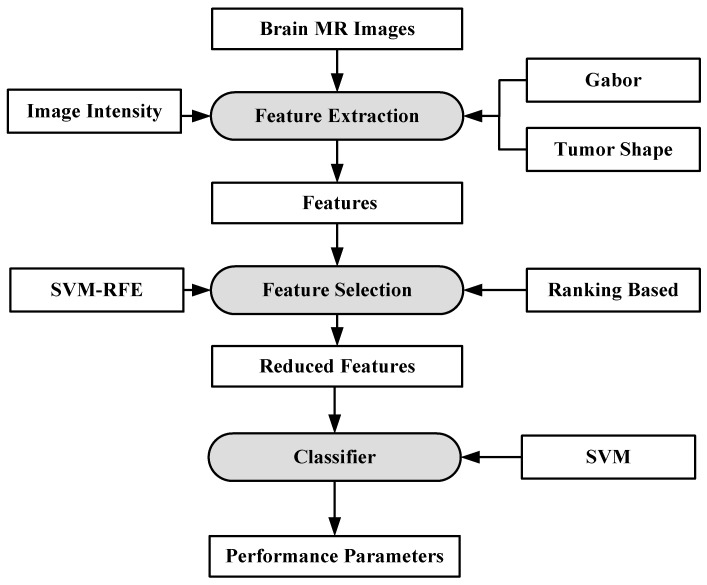
Process model using SVM-based grade estimation method [92].

**Figure 9 cancers-11-00111-f009:**
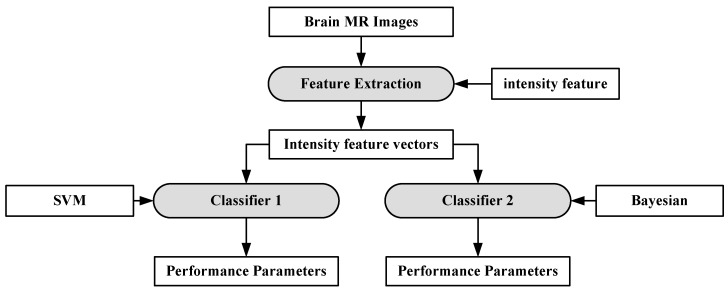
Process model of SVM-based grade estimation method [92].

**Figure 10 cancers-11-00111-f010:**
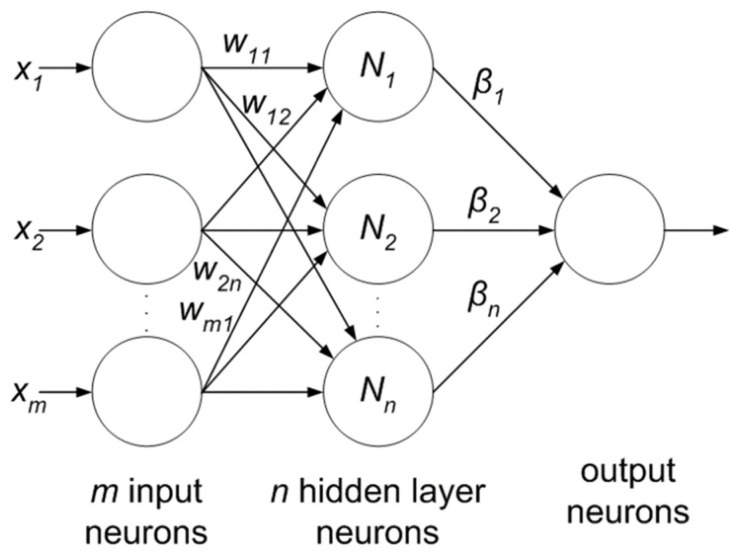
Extreme learning machine.

**Figure 11 cancers-11-00111-f011:**
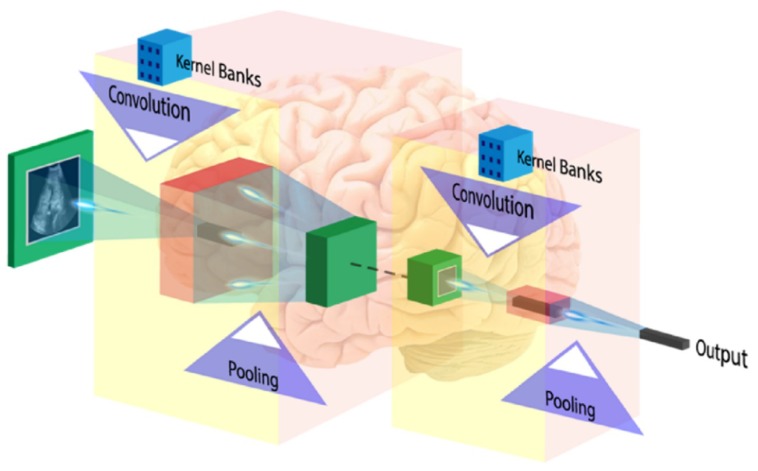
CNN architecture (image courtesy: AtheroPoint^TM^).

**Figure 12 cancers-11-00111-f012:**
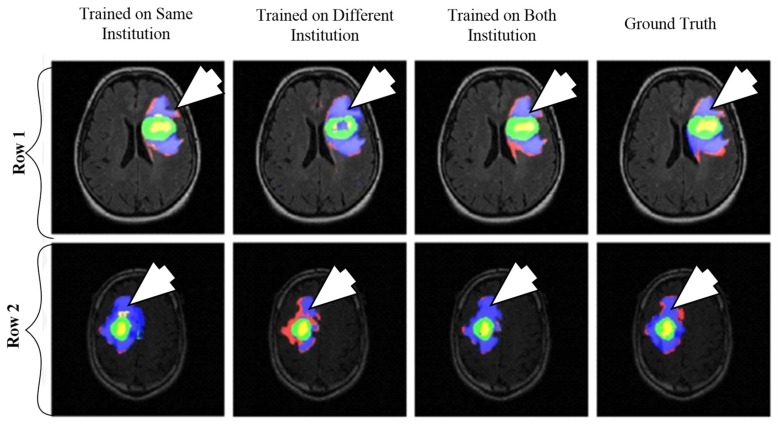
Segmentation results from two different patients. Class1: ground truth; Class 2 (enhancing region): green; Class 3 (necrotic region): yellow, Class 4 (T1abnormality-hypointensity region on T1, excluding enhancing and necrotic regions): red, and Class 5 (FLAIR abnormality excluding classes 2-4): blue (reproduced from [102] with permission).

**Figure 13 cancers-11-00111-f013:**
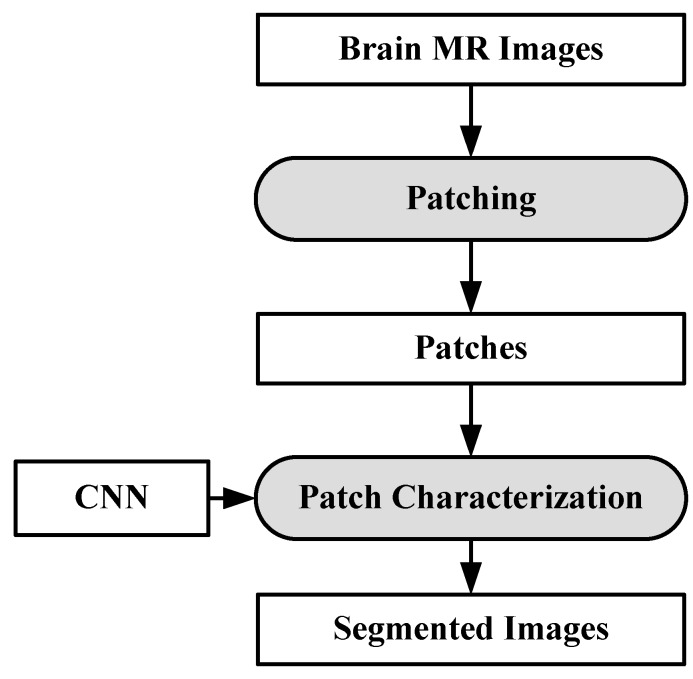
Process model for segmentation [102].

**Figure 14 cancers-11-00111-f014:**
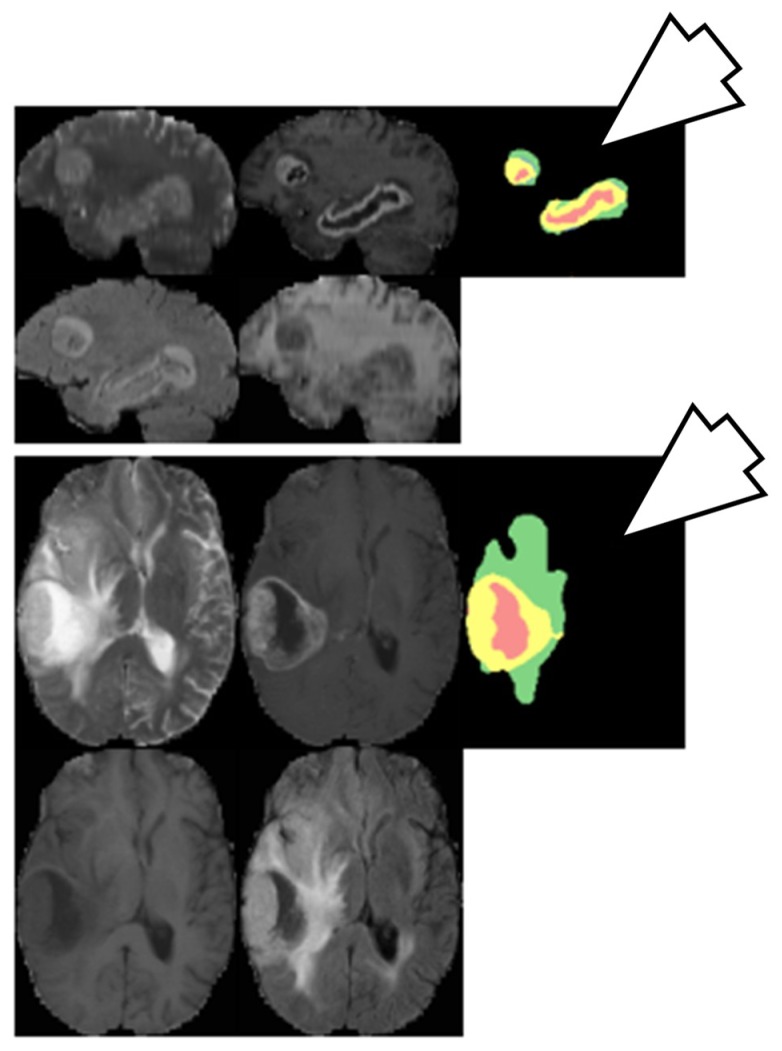
Segmentation results from two different patients. Green: edema, yellow: enhanced tumor, pink: necrosis, blue: non-enhanced tumor (reproduced from [103] with permission).

**Figure 15 cancers-11-00111-f015:**
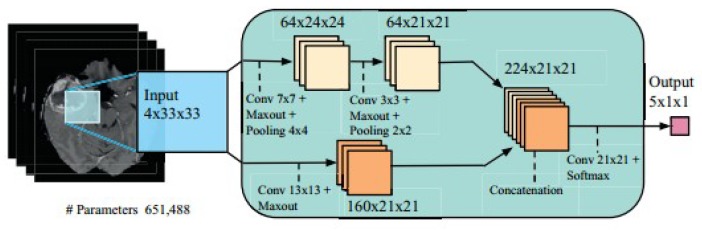
Model Architecture (reproduced from [103] with permission).

**Figure 16 cancers-11-00111-f016:**
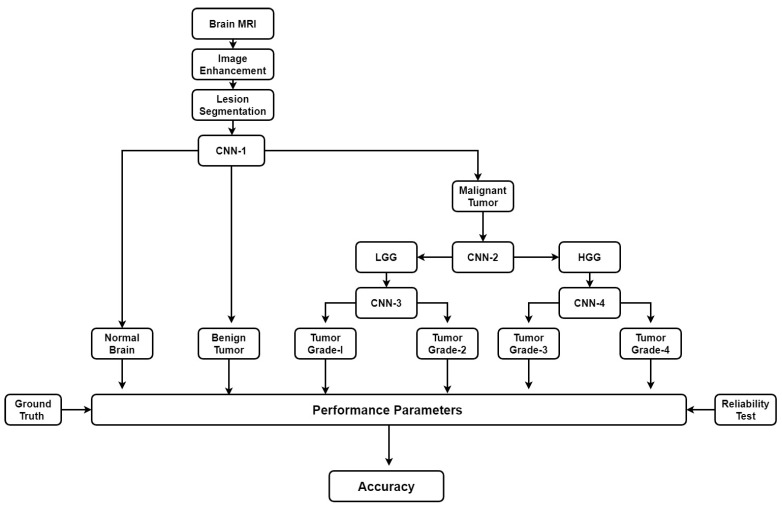
Plausible solution for brain tumor grading.

**Figure 17 cancers-11-00111-f017:**
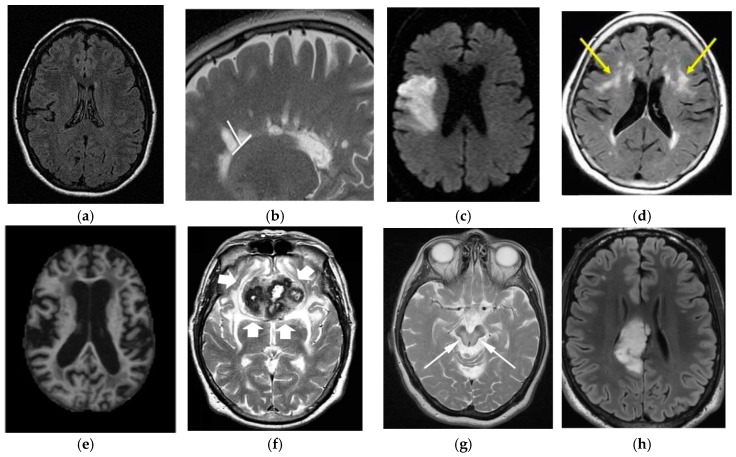
Comparison of brain tumor with other brain disorders (image permission requested from sources). (**a**) Normal Brain [AtheroPoint^TM^]; (**b**) Multiple Sclerosis [113]; (**c**) Stroke [114]; (**d**) Leukoaraiosis [115]; (**e**) Alzheimer’s Disease [116]; (**f**) Parkinson’s Disease [117]; (**g**) Wilson’sDisease [118]; (**h**) Brain Tumor [119].

**Table 1 cancers-11-00111-t001:** Genomics relevance with Brain Tumor, RKT: Receptor Tyrosine Kinase, TP53 (p53): Tumor Protein53, RB1: Retino Blastoma1, EGFR: Epidermal Growth Factor Receptor, PTEN: Phosphatase and Tensin Homolog, IDH1/DH2: Isocitrate Dehydrogenase 1/2, 1p and 19 co-deletion, MGMT: O6-methylguanine DNA methyltransferase, BRAF: B-Raf proto-oncogene, ATRX: The α-thalassemia-mental retardation syndrome X-linked, HGG: High-Grade Gliomas, GBM: Glioblastoma.

Gene Type	Function	Mutation Effect	Relevancy Between Brain Tumor and Genes [Degree of Mutation]
TP53(p53) [26]	DNA repair Initiating Apoptosis	Genetic Instability Reduced Apoptosis Angiogenesis	More relevant to HGG Brain Tumor (80%)
RB1 [26]	Tumor Suppressor	Blocks cell cycle progression Unchecked cell cycle progression	More relevant to GBM Brain Tumor (75%)
EGFR [27]	Trans-Membrane Receptor In (RTK)	Increased Proliferation Increased Tumor Cell Survival	Primary GBM (Approx. 40%)
PTEN [27]	Tumor Suppressor	Increased Cell Proliferation Reduced Cell Death	Primary GBM (15–40%) GBM (up to 80%)
IDH1 and DH2 [28]	Control citric acid cycle	Inhibits the function of enzymes	IDH1 Primary GBM (5%) GBM Grade II-III (70–80%) IDH1 longer survival. IDH2 Relevant to oligodendroglial tumors
1p and 19q [29]	Prognosis of the disease or treatment assessment	Poor prognosis	Oligodendrogliomas (80%) Anaplastic Oligodendrogliomas (60%) Oligoastrocytomas (30–50%) Anaplastic Oligoastrocytomas (20–30%)
MGMT [30]	DNA repair predict patient survival	Cell proliferation	GBM (35–75%)
BRAF [26]	Proto-oncogene	Cell Proliferation Apoptosis	Pilocyticastrocytomas (65–80%) Pleomorphic Xanthoastrocytomas and Gangliogliomas (25%)
ATRX [26]	Deposition of Genomic Repeats.	Genital Anomalies, Hypotonia, Intellectual Disability Mild-To-Moderate Anemia Secondary To α-Thalassemi	Relevent to oligodendroglial

**Table 2 cancers-11-00111-t002:** WHO recommendations for tumor assessment in different editions.

Edition	Year	Recommended Parameters for Tumor Assessment
I	1979	Miotic Activity, Necrosis and Infiltration
II	1993	Immunohistochemistry (IHC)
III	2000	Genetic Profile
IV	2007	Genetic Profile and Histological Variation
V	2016	Molecular Features and Histology

**Table 3 cancers-11-00111-t003:** Overview of some open challenges in digital pathology images analysis worldwide.

Year	Challenges	Reference
2012	ICPR Mitosis Detection Competition	[57]
2012	EM segmentation challenge 2012 2D segmentation of neuronal processes	[58]
2013	MICCAI Grand Challenge on Mitosis Detection	[59]
2014	MICCAI Brain Tumor Digital Pathology Challenge	
2014	MICCAI Brain Tumor Digital Pathology Challenge	
2015	MICCAI Gland Segmentation Challenge Contest	
2016	Tumor Proliferation Assessment Challenge 2016	[60]
2017	CAMELYON17 challenge	[61]
2018	Medical Imaging with Deep Learning (MIDL-2018)	[62]

**Table 4 cancers-11-00111-t004:** Overview open challenges of brain image analysis worldwide.

Challenge	Objective	Modality	Reference
BraTS 2012	Brain Tumor Segmentation	MRI	[64]
BraTS 2013	Brain Tumor Segmentation	MRI	[65]
BraTS 2014	Brain Tumor Segmentation	MRI	[66]
BraTS 2015	Brain Tumor Segmentation	MRI	[67]
BraTS 2016	Quantifying longitudinal changes: evaluate the accuracies of the volumetric changes between any two time points.	MRI	[68]
BraTS 2017	Segmentation of gliomas in pre-operative scans. Prediction of patient overall survival (OS) from pre-operative scans.	MRI	[69]
BraTS 2018	Segmentation of gliomas in pre-operative MRI scans. Prediction of patient overall survival (OS) from pre-operative scans.	MRI	[70]
MICCAI 2018	The segmentation ofgray matter, white matter, cerebrospinal fluid, andother structureson multi-sequence brain MR images with and without (large) pathologies. (large) pathologies on segmentation and volumetry.	MRI	[71]
HC-18	To design an algorithm that can automatically measure the fetal head circumference given a 2D ultrasound image.	Ultrasound Image	[72]

**Table 5 cancers-11-00111-t005:** Overview of Brain Tumor Classification Methods.

Sno	Reference	Tissue Classes	MRI Subtype	Data Size	Feature Processing	Feature Reduction	Architecture for Classification	Highest Performance
1	Sasikala et al. 2008 [63]	N, ABN, B, M	T2W	100, (*N* = 35, *B* = 35, *M* = 30)	DWT	GA	ANN	ACC = 98%; SEN = NA; SPC = NA; AUC = NA
2	Verma et al. 2008 [94]	Neoplasms, edema, and healthy tissue	DWI, B0, FLAIR, T1, and GAD	14 (G-3 = 8, G-4 = 7)			Bayesian, and SVM	ACC = NA; SEN = 91.84; SPC = 99.57; AUC = NA
3	Zacharaki et al. 2009 [92]	Metastasis, meningiomas gliomas (G-2-3) GBM	T1W, T2W, FLAIR, rCBV	102 (Metastasis (24), meningiomas (4), gliomas (G-2) (22), gliomas (G-3) (18), GBM (34))	SVM, RFE	Feature Ranking	LDA, KNN, NL-SVM	ACC = 97.8%; SEN = 100%; SPC = 95%; AUC = 98.6%
4	El-Dahshan et al. 2010 [88]	N, ABN	T2W	60, (*N* = 60, ABN = 10)	DWT	PCA	FP-ANN, KNN	ACC = 98.6%; SEN = 100; SPC = 90; AUC = NA
5	Ryu et al. 2014 [123]	Glioma (G-2,3,4)	DWI, ADC	42 Glioma (G2(N = 8)), G-3 (*N* = 10) and G-4 (*N* = 22))	GLCM		Entropy, Histogram	ACC = 84.4%; SEN = 81.8%; SPC = 90%; AUC = 94.1%
6	Skogenet al. 2016 [105]	LGG (G-2), HGG (G-3-4)	T1W, T2W, FLAIR	95 (LGG = 27 (G-2I) HGG = 68 (G-3 = 34 and G-4 = 34)	Statistical Analysis		Standard Deviation	ACC = 84.4%; SEN = 93%; SPC = 81%; AUC = 91%

GLCM: Gray Level Co-Occurrence Matrix, NL-SVM: Nonlinear SVM, MDF: Most Discriminent Factor, LDA: Linear Discriminant Analysis, ADC: Apparent Diffusion Coefficient, GLCM: Gray Level Co Occurrence Matrix, GA: Genetic Algorithm, DWT: Discrete Wavelet Transform, SVM: Support Vector Machines, RFE: Recursive Feature Elimination, N: Normal, ABN: Abnormal, GBM: Glioblastomas, LGG: Low grade Glioma, HGG: High Grade Glioma, B: Benign, M: Malignant, T1W: T1-Weighted, T2W: T2 Weighted, FLAIR: Fluid-attenuated inversion recovery, rCBV: Relative cerebral blood volume, G: Grade, ANN: Artificial Neural Network, DWT: Discrete Wavelet Transform, FP-ANN: Feedforward, Back Propagation-ANN, ACC: Accuracy, SEN: Sensitivity, SPC: Specificity, AUC: Area Under Curve,. ROC: Receiver Operating Characteristic.

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
