# Peer review of "A Review on a Deep Learning Perspective in Brain Cancer Classification"

_cancers, 2019, doi:10.3390/cancers11010111_

Round 1
Reviewer 1 Report
Review report for “A Review on a Deep Learning Perspective of Brain Cancer Classification" by Gopal S.Tandel, et al.
This review paper summarized the pathophysiology of brain cancer, imaging modalities of brain cancer and automatic computer assisted methods for brain cancer characterization in machine and deep learning paradigm. The manuscript is presented in a clear and systematic way. Therefore, I recommend to accept the manuscript for publication.
Author Response
This review paper summarized the pathophysiology of brain cancer, imaging modalities of brain cancer and automatic computer assisted methods for brain cancer characterization in machine and deep learning paradigm. The manuscript is presented in a clear and systematic way. Therefore, I recommend to accept the manuscript for publication.
Authors: We thank this reviewer for his encouragements.
Reviewer 2 Report
This interesting review gives a good overview on the classification if brain tumors by deep learning based on imaging and histology data. All fiels of the area are discussed and presented comprehensively. The linguistic style is good making the manuscript easy to read and understand. Therefore I strongly advocate publication in "cancers".
Prior to publication I suggest following corrections:
L49: only WHO°I astrocytomas have distinct borders, WHO°II astrocytomas already do not have distinct borders. Please add here that you mean WHO°I astrocytomas only.
L136: I think that you mean here "Epstein-Barr" instead of "Stein-Barr" virus, please correct.
Author Response
This interesting review gives a good overview on the classification if brain tumors by deep learning based on imaging and histology data. All fiels of the area are discussed and presented comprehensively. The linguistic style is good making the manuscript easy to read and understand. Therefore I strongly advocate publication in "cancers".
Prior to publication I suggest following corrections:
(R2-1): L49: only WHO°I astrocytomas have distinct borders, WHO°II astrocytomas already do not have distinct borders. Please add here that you mean WHO°I astrocytomas only.
Authors: We thank the reviewer for correcting us. We have specified “astrocytomas (WHO Grade-I)” at the desirable place (on page 4). Thank you.
(R2-2) L136: I think that you mean here "Epstein-Barr" instead of "Stein-Barr" virus, please correct.
Authors: We thank the reviewer for correcting us. We have replaced "Stein-Barr" to "Epstein-Barr"at the specified location (on page 7). Thank you.
Reviewer 3 Report
In this manuscript authors presents an explanation of the brain tumor causes, its pathophysiology and states the different imaging modalities currently employed to study brain cancer. In addition they provide more explanations about tumor grades and tests performed to evaluate the brain tumors.
Concerning to the review of the deep learning techniques employed to classify brain tumors only a few studies are presented. The other studies presented are related to machine learning approaches that should not be the main focus of this review taking into account the manuscript title. This reviewer has performed a basic search in the literature related to deep learning brain cancer classification and more than 50 papers have been found. As a suggestion, authors should perform a more exhaustive search in different literature databases, including all the relevant papers related with the use of deep learning techniques to classify brain cancer.
Furthermore, the inclusion of section “7. Brain Cancer and other brain disorders”, where other diseases are briefly explained, has no sense for the purpose of this review.
Finally, in the entire manuscript, only studies that employed imaging modalities based on MRI or CT are included. There are other imaging modalities, which where mentioned in section 3, such as spectroscopy or spectral imaging (multispectral or hyperspectral imaging), that also have been researched in the literature as emerging imaging modalities for brain cancer classification.
Author Response
In this manuscript authors presents an explanation of the brain tumor causes, its pathophysiology and states the different imaging modalities currently employed to study brain cancer. In addition they provide more explanations about tumor grades and tests performed to evaluate the brain tumors.
(R3-1) Concerning to the review of the deep learning techniques employed to classify brain tumors only a few studies are presented. The other studies presented are related to machine learning approaches that should not be the main focus of this review taking into account the manuscript title. This reviewer has performed a basic search in the literature related to deep learning brain cancer classification and more than 50 papers have been found. As a suggestion, authors should perform a more exhaustive search in different literature databases, including all the relevant papers related with the use of deep learning techniques to classify brain cancer.
Authors: We thank the reviewer for your kind suggestion and improving our literature quality. In this regard we would like to convey that although there are several imaging modalities, deep learning has not been applied to all types. However, there is a scope of application of deep learning. Our review not only presents the current deep learning applications but also puts forward the possible areas of applications of deep learning. When we are presenting the future scope we are adding the areas where machine learning has already been applied and is currently open to deep learning. In this regard, we have included "Hyperstereoscopy” and “MR Spectroscopy" imaging in section 3 (from page no10-12). This is given as follows:
Hyperstereoscopy Imaging
High-grade tumors invade the surrounding normal tissues, which makes them extremely difficult to differentiate from each other through the naked eyes of surgeon (especially glioma). Incorrect resection leads to reduced survival rate of the brain cancer patients [40,41]. In this case, Hyperspectral imaging (HSI) can be used. HSI is a minimally invasive, non-ionizing sensing technique. HSI contains a wider range of the electromagnetic spectrum compared to normal three channel image of Red, Green and Blue (RGB) type [41], which provides detailed information of tissues in the captured scene [42].
Recently, scientists had proposed a novel visualization system based on HIS, which can assist surgeons to detect the brain tumor boundaries during neurosurgical procedures [40]. This model had used both supervised (SVM and KNN) and unsupervised (K-Means) machine learning techniques to differentiate classes such as normal, cancerous, blood vessels/hyper-vascularized tissue and background in the spectral image. The brain cancer detection algorithm was divided into off-line (training process) and in situ (online) process. In the off-line process, the samples were labeled by the experts and in the in situ process, the HIS images were directly acquired from the patient for real-time image analysis in the operation theater. SVM was adapted for classification during the in situ process to get a supervised classification map, while kNN algorithm was used to find the spatial-spectral classification map. To get the final definitive classification map, image fusion was performed between spatial-spectral classification map (derived from KNN-supervised) and Hierarchical K-Means map (unsupervised strategy). Finally, majority voting (MV) method was used to fuse both images for superior results. For dimensionality reduction, principal component analysis (PCA) algorithm was adapted in the above settings.
In another study utilizing hyperspectral paradigm was [43], where, head and neck cancer classification was done using deep learning (DL) technique. In this study, the author demonstrated that DL techniques have the potential to be used as a real-time tissue classifier (tissue labeling process) using HS images to identify boundaries of the cancerous and non-cancerous tissues during the surgery. A CNN network was proposed consisting of six convolution layers and three fully connected layers to classify three types of classes such as head and neck tissue, squamous-cell carcinoma and thyroid cancer. The database consisted of 50 subjects. The network was trained for 25,000 iterations using a batch size of 250. Performance was evaluated using leave-one-out cross-validation protocol while computing the performance parameters giving the accuracy, sensitivity, specificity as 80%, 81% and 78%, respectively. The CNN strategy was benchmarked against conventional ML methods such as SVM, kNN, logistic regression (LR), decision tree (DT), linear discriminant analysis (LDA) demonstrating its superiority.
MR Spectroscopy
The MRI is able to visualize the anatomical structure of the brain, whereas, Magnetic Resonance spectroscopy (MRS) is able to detect small biochemical changes in the brain. This property is useful for the brain tissue classification in brain tumor, stroke and epilepsy. Here, several metabolites and their products such as amino acids, lactate, lipid, alanin etc., where, the frequency can measured in parts per million (ppm). There are unique metabolic signatures associated with each tumor type and their grades [44], therefore, the neurologist measures the changes between normal and cancerous tissues by the frequency map of the each metabolite’s ppm. In [45], the authors had proposed a deep learning based model for brain tumor diagnosis using MRS imaging techniques. The author had proposed three deep models for brain tumor classification in healthy, low or high grade tissue types. In another study [46], the authors had proposed brain tumors grading method using MR Spectroscopy. The proposed method showed that metabolite values/ratios could provide better classification/grading of brain tumors using, short and long echo time (TE). A machine learning method was proposed by authors in [47] for gliomas classification into benign and malignant types. Features were extracted from MR spectroscopy and then classified using popular ML methods such as SVM, random forest, multilayer perceptron, and locally weighted learning (LWL). The best performance was achieved by random forest giving an AUC of 0.91, while the sensitivity of 86.1% was achieved using LWL-based method.
References:
40. Fabelo, H., Ortega, S., Lazcano, R., Madroñal, D., M Callicó, G., Juárez, E., ...&Piñeiro, J. F. (2018). An intraoperative visualization system using hyperspectral imaging to aid in brain tumor delineation. Sensors, 18(2), 430.DOI: 10.3390/s18020430
41. Petersson, H., Gustafsson, D., & Bergstrom, D. (2016, December). Hyperspectral image analysis using deep learning—a review. In Image Processing Theory Tools and Applications (IPTA), 2016 6th International Conference on (pp. 1-6). IEEE.DOI: 10.1109/IPTA.2016.7820963
42. Lu, G., &Fei, B. (2014). Medical hyperspectral imaging: a review. Journal of biomedical optics, 19(1), 010901.DOI: 10.1117/1.JBO.19.1.010901
43. Halicek, M., Lu, G., Little, J. V., Wang, X., Patel, M., Griffith, C. C., ...&Fei, B. (2017). Deep convolutional neural networks for classifying head and neck cancer using hyperspectral imaging. Journal of biomedical optics, 22(6), 060503.DOI: 10.1117/1.JBO.22.6.060503.
44. Nelson, S. J. (2003). Multivoxel magnetic resonance spectroscopy of brain Tumors1. Molecular cancer therapeutics, 2(5), 497-507.
45. Olliverre, N., Yang, G., Slabaugh, G., Reyes-Aldasoro, C. C., & Alonso, E. (2018, September). Generating Magnetic Resonance Spectroscopy Imaging Data of Brain Tumours from Linear, Non-linear and Deep Learning Models. In International Workshop on Simulation and Synthesis in Medical Imaging (pp. 130-138). Springer, Cham.
46. Hamed, S. A. I., &Ayad, C. E. Grading of Brain Tumors using MR Spectroscopy: Diagnostic value at Short and Long TE.DOI: 10.9790/0853-1610078793
47. Ranjith, G., Parvathy, R., Vikas, V., Chandrasekharan, K., & Nair, S. (2015). Machine learning methods for the classification of gliomas: Initial results using features extracted from MR spectroscopy. The neuroradiology journal, 28(2), 106-111.
(R3-2) Furthermore, the inclusion of section “7. Brain Cancer and other brain disorders”, where other diseases are briefly explained, has no sense for the purpose of this review.
Authors: We thank the reviewer for the suggestion.
While we appreciate the inputs from this reviewer, we would like to most respectfully share with this reviewer that this review though focuses on Brain Cancer, it also provides a perspective how Brain Cancer imaging is different from other brain disorders such as Alzheimer's Disease, Parkinson's Disease etc. At the same time several components of brain disorders are discussed with and without brain cancer conditions.
Therefore most respectfully we would like to specify that this is an important guideline for future researchers in Brain Cancer Imaging. We therefore feel strongly that this adds the value in the case of brain cancer related work. We therefore like to request this reviewer to allow us to keep this small section in the manuscript.
Thank you for the understanding and we appreciate your support.
(R3-3) Finally, in the entire manuscript, only studies that employed imaging modalities based on MRI or CT are included. There are other imaging modalities, which where mentioned in section 3, such as spectroscopy or spectral imaging (multispectral or hyperspectral imaging), that also have been researched in the literature as emerging imaging modalities for brain cancer classification.
Authors: We are thankful to the reviewer for the deep insight and observation coupled with keen interest shown in reviewing this article. In this regard, we have included "Hyperstereoscopy” and “MR Spectroscopy" imaging in section 3 (from page number 10-12). This is given as follows:
Hyperstereoscopy Imaging
High-grade tumors invade the surrounding normal tissues, which makes them extremely difficult to differentiate from each other through the naked eyes of surgeon (especially glioma). Incorrect resection leads to reduced survival rate of the brain cancer patients [40,41]. In this case, Hyperspectral imaging (HSI) can be used. HSI is a minimally invasive, non-ionizing sensing technique. HSI contains a wider range of the electromagnetic spectrum compared to normal three channel image of Red, Green and Blue (RGB) type [41], which provides detailed information of tissues in the captured scene [42].
Recently, scientists had proposed a novel visualization system based on HIS, which can assist surgeons to detect the brain tumor boundaries during neurosurgical procedures [40]. This model had used both supervised (SVM and KNN) and unsupervised (K-Means) machine learning techniques to differentiate classes such as normal, cancerous, blood vessels/hyper-vascularized tissue and background in the spectral image. The brain cancer detection algorithm was divided into off-line (training process) and in situ (online) process. In the off-line process, the samples were labeled by the experts and in the in situ process, the HIS images were directly acquired from the patient for real-time image analysis in the operation theater. SVM was adapted for classification during the in situ process to get a supervised classification map, while kNN algorithm was used to find the spatial-spectral classification map. To get the final definitive classification map, image fusion was performed between spatial-spectral classification map (derived from KNN-supervised) and Hierarchical K-Means map (unsupervised strategy). Finally, majority voting (MV) method was used to fuse both images for superior results. For dimensionality reduction, principal component analysis (PCA) algorithm was adapted in the above settings.
In another study utilizing hyperspectral paradigm was [43], where, head and neck cancer classification was done using deep learning (DL) technique. In this study, the author demonstrated that DL techniques have the potential to be used as a real-time tissue classifier (tissue labeling process) using HS images to identify boundaries of the cancerous and non-cancerous tissues during the surgery. A CNN network was proposed consisting of six convolution layers and three fully connected layers to classify three types of classes such as head and neck tissue, squamous-cell carcinoma and thyroid cancer. The database consisted of 50 subjects. The network was trained for 25,000 iterations using a batch size of 250. Performance was evaluated using leave-one-out cross-validation protocol while computing the performance parameters giving the accuracy, sensitivity, specificity as 80%, 81% and 78%, respectively. The CNN strategy was benchmarked against conventional ML methods such as SVM, kNN, logistic regression (LR), decision tree (DT), linear discriminant analysis (LDA) demonstrating its superiority.
MR Spectroscopy
The MRI is able to visualize the anatomical structure of the brain, whereas, Magnetic Resonance spectroscopy (MRS) is able to detect small biochemical changes in the brain. This property is useful for the brain tissue classification in brain tumor, stroke and epilepsy. Here, several metabolites and their products such as amino acids, lactate, lipid, alanin etc., where, the frequency can measured in parts per million (ppm). There are unique metabolic signatures associated with each tumor type and their grades [44], therefore, the neurologist measures the changes between normal and cancerous tissues by the frequency map of the each metabolite’s ppm. In [45], the authors had proposed a deep learning based model for brain tumor diagnosis using MRS imaging techniques. The author had proposed three deep models for brain tumor classification in healthy, low or high grade tissue types. In another study [46], the authors had proposed brain tumors grading method using MR Spectroscopy. The proposed method showed that metabolite values/ratios could provide better classification/grading of brain tumors using, short and long echo time (TE). A machine learning method was proposed by authors in [47] for gliomas classification into benign and malignant types. Features were extracted from MR spectroscopy and then classified using popular ML methods such as SVM, random forest, multilayer perceptron, and locally weighted learning (LWL). The best performance was achieved by random forest giving an AUC of 0.91, while the sensitivity of 86.1% was achieved using LWL-based method.
References:
40. Fabelo, H., Ortega, S., Lazcano, R., Madroñal, D., M Callicó, G., Juárez, E., ...&Piñeiro, J. F. (2018). An intraoperative visualization system using hyperspectral imaging to aid in brain tumor delineation. Sensors, 18(2), 430.DOI: 10.3390/s18020430
41. Petersson, H., Gustafsson, D., & Bergstrom, D. (2016, December). Hyperspectral image analysis using deep learning—a review. In Image Processing Theory Tools and Applications (IPTA), 2016 6th International Conference on (pp. 1-6). IEEE.DOI: 10.1109/IPTA.2016.7820963
42. Lu, G., &Fei, B. (2014). Medical hyperspectral imaging: a review. Journal of biomedical optics, 19(1), 010901.DOI: 10.1117/1.JBO.19.1.010901
43. Halicek, M., Lu, G., Little, J. V., Wang, X., Patel, M., Griffith, C. C., ...&Fei, B. (2017). Deep convolutional neural networks for classifying head and neck cancer using hyperspectral imaging. Journal of biomedical optics, 22(6), 060503.DOI: 10.1117/1.JBO.22.6.060503.
44. Nelson, S. J. (2003). Multivoxel magnetic resonance spectroscopy of brain Tumors1. Molecular cancer therapeutics, 2(5), 497-507.
45. Olliverre, N., Yang, G., Slabaugh, G., Reyes-Aldasoro, C. C., & Alonso, E. (2018, September). Generating Magnetic Resonance Spectroscopy Imaging Data of Brain Tumours from Linear, Non-linear and Deep Learning Models. In International Workshop on Simulation and Synthesis in Medical Imaging (pp. 130-138). Springer, Cham.
46. Hamed, S. A. I., &Ayad, C. E. Grading of Brain Tumors using MR Spectroscopy: Diagnostic value at Short and Long TE.DOI: 10.9790/0853-1610078793
47. Ranjith, G., Parvathy, R., Vikas, V., Chandrasekharan, K., & Nair, S. (2015). Machine learning methods for the classification of gliomas: Initial results using features extracted from MR spectroscopy. The neuroradiology journal, 28(2), 106-111.
Authors: We have also added a note on biomarkers for early cancer detection and future applications of machine and deep learning (page no 33). It is given as:
“Various tests have been suggested for diagnosing brain cancer: (a) including the one stated earlier in the section of imaging modalities, such as MRI, MRS, CT, etc., and (b) laboratory sampling of brain tumor i.e., biopsy. The inclusion of intelligence-based techniques such as ML or DL for imaging modalities are very likely to increase the effectiveness of the diagnosis and enhance the radiologists' capability towards accurate diagnosis for brain cancer in a timely manner. In addition to the computer-aided diagnosis using imaging modalities and biopsy methodologies, spread of cancer in the nervous system can be detected using a sample of cerebrospinal fluid from the spinal cord. This technique is called lumbar puncture or spinal tap [100]. In this methodology, several biomarkers related to brain tumor were detected [101]. In addition, molecular tests on brain tumor sample can be carried out to identify specific genes, proteins, and cells related to the particular tumor. Doctors can look into these biomarkers to assess the grade, type of tumor and decide treatment options. Further, examining these biomarkers can help in early treatment before the symptoms begin. Inclusion of ML and DL techniques in assessing these biomarkers can lead to accurate diagnosis that can save both time and cost, proving to be more economical”.
References:
100. Quincke, H. I. "Lumbar puncture." Diseases of the nervous system, Church A.(Ed), Appleton, New York (1909): 223.
101. Lynch, H.T., Lynch, J.F., Shaw, T.G. and Lubiński, J., 2003. HNPCC (Lynch Syndrome): Differential Diagnosis, Molecular Genetics and Management-a Review. Hereditary Cancer in Clinical Practice, 1(1), p.7.
Round 2
Reviewer 3 Report
Authors have addressed the previous comments including more references about other works presented in the literature related with brain cancer and machine/deep learning analysis. For this reason I recommend the publication of the manuscript in the present form.
Only two minor typo mistakes:
"Recently, scientists had proposed a novel visualization system based on HIS..."
"...the HIS images were directly acquired...
"HIS" should be "HSI"